Robust multi-view locality preserving regression embedding

http://orcid.org/0000-0001-7813-6512 Jing Ling 1 2 3
Li Yi 2
Zhang Hongjie 4 zhanghj@tiangong.edu.cn
1 College of Science, China Agricultural University , Beijing , China
2 College of Information and Electrical Engineering, China Agricultural University , Beijing , China
3 National Innovation Center for Digital Fishery, China Agricultural University , Beijing , China
4 School of Mathematical Sciences, Tiangong University , Tianjin , China
Alatas Bilal
Electronic publication date: 2024 Dec 20
Publication date: 2024
Volume: 10
Electronic Location ID: e2619
Received 2024 Sep 16; Accepted 2024 Nov 29
Copyright: © 2024 Jing et al.
Copyright year: 2024
Copyright holder: Jing et al.
License: This is an open access article distributed under the terms of the Creative Commons Attribution License, which permits unrestricted use, distribution, reproduction and adaptation in any medium and for any purpose provided that it is properly attributed. For attribution, the original author(s), title, publication source (PeerJ Computer Science) and either DOI or URL of the article must be cited.
License URL: https://creativecommons.org/licenses/by/4.0/

Keywords: Multi-view learning, Feature extraction, Graph embedding, Regression embedding

Funding: National Natural Science Foundation of China 62076244, 12071024 Beijing Digital Agriculture Innovation Consortium Project BAIC10-2023 National Shrimp and Crab Industry Technical System Construction Project 2022 CARS-48 The work was supported by the National Natural Science Foundation of China (Nos. 62076244, 12071024), the Beijing Digital Agriculture Innovation Consortium Project (BAIC10-2023), and the National Shrimp and Crab Industry Technical System Construction Project 2022 (No. CARS-48). The funders had no role in study design, data collection and analysis, decision to publish, or preparation of the manuscript.

==============================
Feature extraction research has witnessed significant advancements in recent decades, particularly with single-view graph embedding (GE) methods that demonstrate clear advantages by incorporating structural information. However, multi-view data includes descriptions from various perspectives or sensors, offering richer and more comprehensive information compared to single-view data. Research interest in multi-view feature extraction is steadily increasing. Hence, there is a pressing need for a comprehensive framework that extends single-view methods, especially effective GE methods, into multi-view approaches. This article proposes three innovative multi-view feature extraction frameworks based on regression embedding. These frameworks extend single-view GE methods to the multi-view scenario. Our approach meticulously considers the consistency and complementarity of multi-view data, placing strong emphasis on robustness to noisy datasets. Additionally, the use of non-linear shared embedding helps prevent the loss of essential information that may occur with linear projection techniques. Through numerical experiments, we validate the effectiveness and robustness of our proposed frameworks on both real and noisy datasets.

Introduction

The continuous advancement of information technology has significantly enhanced the capabilities of data collection, transmission, storage, processing, and utilization. In theory, a higher numerical feature dimension within a given image sample allows for the inclusion of more valuable information for identification and classification (Braik et al., 2024; Li et al., 2024; Huang et al., 2023). However, practical applications reveal that an excessively high feature dimension introduces a range of challenges in data processing (Jain, Duin & Mao, 2000). These challenges include issues related to data storage and computational complexity, a high redundancy among data features, and the occurrence of the Hughes phenomenon (Hughes, 1968). To address the drawbacks of high dimensionality, feature extraction technology offers a solution that reduces computational costs and improves the efficiency of machine learning. While multi-view feature extraction may somewhat lag behind deep learning algorithms in terms of performance, it provides notable advantages in interpretability and compatibility with various hardware types, including central processing units (CPUs), graphic processing units (GPUs), and digital signal processor (DSPs). These strengths contribute to its continued relevance as an important research focus. Therefore, advancing traditional multi-view feature extraction methods to more effectively extract discriminative features remains a significant pursuit.

The categorization of feature extraction methods into single-view feature extraction (Nie et al., 2023; Li et al., 2022; Zhang et al., 2022a) and multi-view feature extraction (Wei et al., 2023; Wong et al., 2022; Qiang et al., 2022) is primarily based on the type of data samples. In single-view feature extraction, principal component analysis (PCA) (Oh & Kwak, 2016) is a commonly used technique. PCA is an unsupervised approach that aims to find the subspace with the highest sample variance, making it suitable for various subsequent tasks. However, PCA is less effective when dealing with nonlinear data. As manifold learning techniques have evolved, various nonlinear feature extraction methods like locality preserving projections (LPP) (He & Niyogi, 2003), neighborhood preserving embedding (NPE) (He et al., 2005), and isometric projection (Cai, He & Han, 2007) have been introduced to address the limitations of linear methods like PCA. These methods aim to retain the original structures in the subspace but may require different prerequisites. For example, LPP relies on the upfront generation of a neighbor graph for the original data. NPE assumes that the linear reconstruction relationship of the original samples can be maintained between the samples and neighboring samples in embedding space. In contrast, supervised feature extraction methods leverage sample labels to provide more discriminant information. Linear discriminant analysis (LDA) (Belhumeur, Hespanha & Kriegman, 1997) seeks to find a transformation that minimizes within-class scatter and maximizes between-class scatter. However, like PCA, LDA is inherently linear, which might not yield optimal results when samples within a class form separate clusters. To overcome this, researchers have introduced methods like local fisher discriminant analysis (Sugiyama, 2007) and marginal fisher analysis (Yan et al., 2007) to consider distinct local structures. Additionally, sparsity-preserving discriminant projections (Qiao, Chen & Tan, 2010) have been developed based on sparse reconstruction to maintain sparse reconstruction coefficients in the subspace. Despite the distinct motivations behind these algorithms, Yan et al. (2007) have presented a general framework known as graph embedding (GE) that unifies the above unsupervised and supervised methods within a common framework. In this framework, each algorithm is designed to preserve a specific structure within a dataset, improving feature extraction performance and providing a platform for developing new dimensionality reduction algorithms based on various graph structures.

Furthermore, multi-view data typically offers a wealth of complementary information compared to single-view data (Li et al., 2024; Feng & Wang, 2023; Liu et al., 2023). Multi-view feature extraction, which amalgamates relationships between different views to enhance model performance, has found extensive application across various domains (Zhang et al., 2022b; Liang et al., 2022; Zheng et al., 2021). Here, the most straightforward approach involves consolidating all features into an extended vector and then applying a single-view method for feature extraction. However, this method tends to be computationally expensive and doesn’t effectively consider the information related to consistency and complementarity among multiple views. Canonical correlation analysis (CCA) (Hardoon, Szedmák & Shawe-Taylor, 2004) is a prominent unsupervised multi-view technique aimed at uncovering consistency information. It achieves this by optimizing two projection matrices to maximize the correlation between different representations of the same embedding sample from two distinct views. In the realm of single-view feature extraction techniques, which prioritize structure preservation, variations of CCA have been introduced. Locality-preserving CCA (LPCCA) (Sun & Chen, 2007), including its variant, a new LPCCA (Wang & Zhang, 2013), focuses on retaining local neighbor relationships during the computation of the projection matrices. It is essential to note that manual neighbor selection in these methods may introduce inaccuracies. To overcome this shortcoming, canonical sparse cross-view correlation analysis (Zu & Zhang, 2016) leverages sparse representation to unveil local geometric structures without manual intervention. Additionally, Rupnik & Shawe-Taylor (2010) introduced multi-view CCA (MCCA), an extension of CCA designed for handling data with more than two views. Based on this, discriminative MCCA (Gao et al., 2018) considers label information and falls under supervised multi-view feature extraction. Beyond CCA-based models, researchers have proposed the multi-view uncorrelated discriminant analysis method (MvDA) Shu et al. (2019), drawing inspiration from single-view feature extraction techniques like LDA and PCA. MvDA strives to identify a shared subspace while maintaining intra-class compactness and inter-class separation. Wei et al. (2023) introduced an entropy-weighted non-negative matrix factorization (EWNMF) that assigns weights to each data attribute, placing emphasis on their importance.

While single-view feature extraction research has made significant progress over the past few decades, it has recently reached a plateau. The widespread availability of multi-view data across various domains and the remarkable performance of multi-view learning have shifted the focus of current research towards multi-view feature extraction. However, due to the specific characteristics of multi-view data, single-view feature extraction methods cannot be directly applied to multi-view data, which will lead to the waste of the long-developed single-view feature extraction methods. Therefore, there is an urgent need for a comprehensive framework that extends single-view methods, especially the effective GE methods, into multi-view approaches. This extension should fully consider the unique attributes of multi-view data. In response to this need, we propose the multi-view regression embedding frameworks. Our frameworks extend single-view GE methods into a multi-view context, carefully considering the consistency and complementarity of multi-view data. Specifically, our frameworks apply GE to shared embeddings from multiple views, effectively capturing the consistency information among these views. This shared embedding is obtained through non-linear projections, derived by fitting linear projections of different views using regression techniques. The use of non-linear shared embedding helps prevent the loss of essential information that can occur with linear projection techniques. Furthermore, our frameworks introduce adaptive weights to the GEs and the regression techniques of different views, effectively leveraging the complementary information across multiple views. Additionally, by imposing distinct norm constraints on the projection matrices, our framework enhances its robustness to noisy data.

The main contributions of this article can be summarized as follows: The multi-view regression embedding frameworks are proposed to extend the effectiveness single-view GE methods to the multi-view feature extraction.

The frameworks comprehensively considers the consistency and complementarity of multi-view data, enhancing robustness through the incorporation of norm constraints.

The framework utilizes nonlinear shared embedding to prevent the loss of crucial information that can occur with linear projections.

Various multi-view feature extraction models are constructed within the framework, and their performance is confirmed through real dataset evaluations.

The structure of this article is outlined as follows: ‘Related Work’ briefly describes the GE framework and two specific methods PCA and LDA under this framework. ‘Proposed Method’ presents the multi-view feature extraction framework and expounds on two extended frameworks. ‘Optimization strategy’ details the algorithm design of the feature extraction framework. ‘Experiments’ carries out experiments on three datasets to assess the proposed approach. ‘Conclusion’ concludes the article by presenting a summary and analysis of the entire work.

Related work

Single-view feature extraction problem: Given a training sample set X=[x1,x2,…,xn]∈Rd×n, where n and d are the numbers of samples and features, respectively. In the supervised case, labels for samples are provided, and the label of xi is defined as ci, i=1,2,…,n. The goal of feature extraction is to find a projection matrix P∈Rd×m, where m≪d. The embedding space Y=[y1,y2,…,yn] ∈Rm×n is represented by Y=PTX.

GE

The single-view feature extraction methods based on sample structures can be unified under the GE framework. In GE, a similarity matrix WS∈Rn×n is constructed using distinct structural information of orignal samples, where WSij represents the similarity of samples xi and xj. This framework entails learning the projection matrix P by embedding the graph structures into a regression model. The GE framework can be summarized as follows:

(1) minPG(Y)=∑i,j=1n∥yi−yj∥22Wijss.t. YHYT=I or PTP=I,

where I is an identity matrix. H represents the constraint matrix designed to prevent a trivial solution for the objective function. Therefore, the specific models in the GE framework are determined by the specific constraint and the matrices WS and H. The above objective function can be further transformed into:

(2) G(Y)=YTLY,

where L is the Laplacian matrix of WS.

PCA

PCA is an unsupervised GE model that aims to identify a set of orthogonal axes representing directions with the highest variance in the original data. It then projects the data onto the first m of these axes, selectively retaining the feature dimensions that contain the majority of the variance while disregarding dimensions with nearly zero variance, resulting in the reduction of data feature dimensionality. The constraint of PCA is given by:

(3) PTP=I.

The similarity matrix WS is defined as follows:

(4) WSij={1n,ifi≠j0,otherwise.

Thus, the Laplacian matrix of WS is computed as follows:

(5) L=I−1n11T,

where 1 is an n-dimensional column vector of all 1.

LDA

LDA is a supervised GE model that aims to ensure that during the projection process, the data in the new subspace has the maximum inter-class distance and the minimum intra-class distance. This is done to enhance the separability of the data after projection, aiming to improve classification or clustering performance. The constraint of LDA is given by:

(6) YHYT=I,

where

(7) H=I−1n11T.

The similarity matrix WS is defined as follows:

(8) WijS={1ni,ifci = cj0,otherwise.

Thus, the Laplacian matrix of WS is computed as follows:

(9) L=I−∑c=1nc1nc1c1cT,

nc is the number of samples in c th class. 1c is an n-dimensional column vector with 1c(i)=1 if c=ci; 0 otherwise.

Proposed method

This section proposes three multi-view feature extraction frameworks based on regression embedding, namely multi-view regression embedding (MRE), multi-view locally preserved regression embedding (MLPRE), robust multi-view locally preserved regression embedding (RMLPRE). Specific multi-view feature extraction models are devised by integrating traditional single-view PCA and LDA into these frameworks. An illustrative overview of these frameworks construction process is provided in Fig. 1.

Figure 1 Process of constructing MRE and expanding frameworks.

Multi-view feature extraction problem can be defined as follows: Let v be the number of multi-view data sources, and consider multi-view data samples X(i)=[x1(i),x2(i),…,xn(i)]∈Rd(i)×n. Here, d(i) represents the dimensionality of the feature space for the i-th view, and n is the total number of data samples. The goal of multi-view feature extraction is to learn projection matrices P(i)∈Rd(i)×m, where m≪d(i). These projection matrices aim to transform the original data into embedding representations Y(i)=P(i)TX(i)∈Rm×n.

MRE

To comprehensively account for the consistency and complementarity information among views while extending the single-view GE framework to multi-view feature extraction, we apply GE by incorporating the structural information from distinct views into shared embedding. Additionally, we introduce adaptive weights to the regression terms and the GEs in distinct views. The optimization problem is as follows:

(10) minα,Y,P(i)∑i=1vαir[G(Y,X(i))+12‖P(i)TX(i)−Y‖F2], s.t. αi≥0,∑i=1vαi=1,YHYT=IorPT(i)P(i)=I,i=1,...,v,

where

(11) G(Y,X(i))=YL(i)YT,i=1,…,v,

The parameter r is a positive value, and α=[α1,…,αv]. L(i) represents the Laplacian matrix of the similarity matrix in i th view. In this context, the second term of the objective function employs regression techniques to fit linear projections of different views to the shared embedding Y. This fitting process yields fitting residuals Yr(i), which has the following representation:

(12) Y=P(i)TX(i)+Yr(i),i=1,2,…,v,

This implies that the shared embedding Y is a nonlinear embedding due to this regression-based transformation. Therefore, this approach effectively prevents the loss of crucial information associated with linear projection methods.

To avoid over-fitting and prevent degenerate trivial solutions, we introduced a regularization term for the norm of the projection matrices ‖P(i)T‖F2. Additionally, we introduced a structured regularization term ‖Yi−Y¯‖F2 to maximize the overall divergence of the samples after projection, thereby retaining more complete information from each view. Here, Y¯ is calculated as:

(13) Y¯=1vY⋅1⋅1T.

The optimization problem is as follows:

(14) minαir,Y,P(i)∑i=1vαir[G(Y,X(i))+12‖P(i)TX(i)−Y‖F2+γ2‖P(i)T‖F2]‖Yi−Y―‖F2, s.t. αi≥0,∑i=1vαi=1.

In Eq. (14), the constraint YHYT=I or PT(i)P(i)=I, i=1,…,v has been replaced by the regularization term ||P(i)T||F2 in the objective function. Therefore, it can be omitted from the constraints.

MLPRE

To further explore the consistency information, we leverage the shared k-nearest neighbor structure across all views to constrain the shared embedding. The optimization problem can be stated as follows:

(15) minαi,Y,P(i)∑i=1vαir[G(Y,X(i))+12‖P(i)TX(i)−Y‖F2+γ2‖P(i)T‖F2+λ2∑j,k=1nWjkLP(i)‖Yj−Yk‖F2]‖Y−Y―‖F2,s.t. αi≥0,∑i=1vαi=1,

where

(16) WjkLP(i)={exp(−‖xj(i)−xk(i)‖22t),xi  and  xj  arek-neighbors in all views.0,otherwise

Note that GE uses the graph structure under each view to constrain the shared embedding Y separately, whereas here the graph structure common to all views is used to constrain the shared embedding Y. Thus it further facilitates the exploration of the consistency information, and thus also weakens the detrimental effect of the inaccuracy of the graph structure in a particular view due to the noise and redundant features.

RMLPRE

To enhance robustness against noisy data, we employ distinct norms for the projection matrices within the proposed framework. The resulting model is expressed as follows:

(17) minαi,Y,P(i)∑i=1vαir[G(Y,X(i))+12‖P(i)TX(i)−Y‖F2+γ2‖P(i)T‖β+λ2∑j,k=1nWjkLP(i)‖Yj−Yk‖F2]‖Y−Y―‖F2, s.t. αi≥0,∑i=1vαi=1.

In Eq. (17), the symbol ‖⋅‖β denotes a particular matrix norm, for example L1-norm or L2,1-norm. Notably, research based on L1-norm and L2,1-norm has shown that these norms exhibit improved performance in the presence of outliers compared to F-norm-based methods. Moreover, L2,1-norm-based techniques are often easier to solve than their L1-norm counterparts.

Framework application

In this section, we extend the single-view PCA and LDA into six multi-view feature extraction models by the three frameworks proposed above, respectively, namely pcMRE, pcMLPRE, pcRMLPRE, daMRE, daMLPRE, and daRMLPRE.

To derive the optimization problems for pcMRE, pcMLPRE, and pcRMLPRE, the Laplacian matrix for each view is calculated as follows:

(18) L(i)=I−1n1⋅1T,i=1,…,v,

with the L2,1-norm regularization applied to objective function of pcRMLPRE. The optimization problems can be formulated as follows:

1) pcMRE:

(19) minαi,Y,P(i)∑i=1vαir[tr(Y(I−1n1⋅1T)YT)+12‖P(i)TX(i)−Y‖F2+γ2‖P(i)T‖F2]‖Y−Y―‖F2,s.t. αi≥0,∑i=1vαi=1.

2) pcMLPRE:

(20) minαi,Y,P(i)∑i=1vαir[tr(Y(I−1n1⋅1T)YT)+12‖P(i)TX(i)−Y‖F2+γ2‖P(i)T‖F2+λ2∑j,k=1nWjkLP(i)‖Yj−Yk‖F2]‖Y−Y―‖F2,s.t. αi≥0,∑i=1vαi=1.

3) pcRMLPRE:

(21) minαi,Y,P(i)∑i=1vαir[tr(Y(I−1n1⋅1T)YT)+12‖P(i)TX(i)−Y‖F2+γ2‖P(i)T‖2,1+λ2∑j,k=1nWjkLP(i)‖Yj−Yk‖F2]‖Y−Y¯‖F2,s.t. αi≥0,∑i=1vαi=1.

To derive the optimization problems for daMRE, daMLPRE and daRMLPRE, the Laplacian matrix for each view is calculated as follows:

(22) L(i)=I−∑c=1nc1nc1c⋅1cT,i=1,…,v,

with the L2,1-norm regularization applied to objective function of daRMLPRE. The optimization problems can be formulated as follows:

1) daMRE:

(23) minαi,Y,P(i)∑i=1vαir[tr(Y(I−∑c=1nc1nc1c⋅1cT)YT)+12‖P(i)TX(i)−Y‖F2+γ2‖P(i)T‖F2]‖Y−Y―‖F2,s.t. αi≥0,∑i=1vαi=1.

2) daMLPRE:

(24) minαi,Y,P(i)

(25) ∑i=1vαir[tr(Y(I−∑c=1nc1nc1c⋅1cT)YT)+12‖P(i)TX(i)−Y‖F2+γ2‖P(i)T‖F2+λ2∑j,k=1nWjkLP(i)‖Yj−Yk‖F2]‖Y−Y¯‖F2,s.t. αi≥0,∑i=1vαi=1.

3) daRMLPRE:

(26) minαi,Y,P(i)

(27) ∑i=1vαir[tr(Y(I−∑c=1nc1nc1c⋅1cT)YT)+12‖P(i)TX(i)−Y‖F2+γ2‖P(i)T‖2,1+λ2∑j,k=1nWjkLP(i)‖Yj−Yk‖F2]‖Y−Y¯‖F2,s.t. αi≥0,∑i=1vαi=1.

In summary, the integration of the Laplacian matrices L(i), i=1,2,…,v from PCA and LDA into our proposed frameworks allows for the development of tailored methods. Moreover, the Laplacian matrices from other single-view graph embedding methods can also be seamlessly incorporated into these frameworks. As a result, our frameworks effectively bridge the gap, expanding single-view graph embedding techniques into the domain of multi-view feature extraction and, in turn, propelling the progress of multi-view learning.

Optimization strategy

We designed a simplified iterative algorithm to replace the traditional alternating iterative algorithm to reduce the cost of each iteration.

Optimization of MRE and MLPRE

Let Eq. (15) be f(Y,P(i),αi):

(28) ∂f∂P(i)T=g(P(i)TX(i)X(i)T+γP(i)T),

where g is independent of P(i)T. Let the partial derivative be 0:

(29) P(i)T=YX(i)T(X(i)X(i)T+γI)−1.

Substituting Eq. (29) into Eq. (15), we have

(30) minαi,Y,P(i)∑i=1vαir[G(Y,X(i))+12‖P(i)TX(i)−Y‖F2+γ2‖P(i)T‖F2+λ2∑j,k=1nWjkLP(i)‖Yj−Yk‖F2]‖Y−Y¯‖F2=>minαi,Y∑i=1vαir[G(Y,X(i))+tr[Y(M(i))YT]]tr[YNYT],

where

(31) M(i)=I−X(i)T(X(i)X(i)T+γI)−1X(i)+λαir(D−W)

and

(32) N=I−1n1⋅1T.

If G(Y,X(i)) can be written as tr[[Y(L(i))YT], we have

(33) minαi,Y∑i=1vαir[G(Y,X(i))+tr[Y(M(i))YT]]tr[YNYT]=>minαi,Ytr[Y(∑i=1vαir(L(i)+M(i)))YT]tr[YNYT]=>minαi,Ytr[Y(∑i=1vαir(L(i)+M(i)))YT],s.t.YNYT=I.

Step 1: Updating Y while fixing αi. By constructing the multivariate Lagrange function and set the partial derivative of YT to 0, the optimization problem Eq. (33) can be solved by the following generalized eigenvalue problem

(34) (∑i=1vαir(L(i)+M(i)))YT=μNYT.

Step 2: Updating αi while fixing Y. The optimal solution of problem Eq. (33) can be calculated as:

(35) αi=[1/tr(Y(L(i)+M(i)))YT)](1/(r−1))∑i=1v[1/tr(Y(L(i)+M(i)))YT)](1/(r−1)).

We alternatively update Y and αi until convergence. P(i)T can be solved by Formula (29). The complete procedures are described in Algorithm 1.

Algorithm 1 Algorithm of MRE and MLPRE.

Input: Given a set of multi-view datasets X(i)∈Rd(v)×n, the number of iterations T, the paramenter r, T and γ, and the embedded space dimensions m.	
Initialization: αi=1/v;	
Compute M(i) by Eq. (31);	
Compute N by Eq. (32);	
  for t=1:T do	
    Compute Y according to Eq. (34);	
    Update αi according to Eq. (35);	
    Compute P(i) according to Eq. (29);	
  end for	
  Output: The contribution weight αi, and the projection matrix P(i).	

Optimization of RMLPRE

Substituting Eq. (29) into Eq. (17), Eq. (17) can be further written as follows:

If G(Y,X(i)) can be written as tr[Y(L(i))YT], we have

(36) minαi,Y,P(i)∑i=1vαir[G(Y,X(i))+12‖P(i)TX(i)−Y‖F2+γ2‖P(i)T‖2,1+λ∑i,j=1n12WijLP‖Yi−Yj‖F2]‖Y−Y¯‖F2=>minαi,Y,P(i)∑i=1vαir[12‖P(i)TX(i)−Y‖F2+γ2‖P(i)T‖2,1+tr[Y[L(i)+λαi(D(i)−WLP(i))]YT]],s.t. αi≥0,∑i=1vαi=1,YNYT=I,

where N=I−1n1⋅1T.

Step 1: Updating P(i) while fixing αi and Y. Problem Eq. (36) can be derived by minimizing the following function

(37) f(P(i))=αir[12‖P(i)TX(i)−Y‖F2+γ2‖P(i)T‖2,1].

Take the derivative of f(P(i)) and set it to zero

(38) P(i)T=YX(i)T(X(i)X(i)T+12γW(i))−1,

where

(39) W(i)=[1‖P(i)1T‖2 ⋱ 1‖P(i)diT‖2],

and di is the dimension of the i th view.

Step 2: Updating Y while fixing αi and P(i). Problem Eq. (36) is equivalent to

(40) minαi,Y,P(i)∑i=1vαir[12‖P(i)TX(i)−Y‖F2+tr[Y[L(i)+λαir(D(i)−WLP(i))]YT]],s.t.YNYT=I.

Substituting Eq. (38) into Eq. (40), the optimization equation can be solved by the following generalized eigenvalue problem

(41) [∑i=1vαirG(i)+λ(D−WLP)]YT=μNYT,

where G(i)=X(i)T[(X(i)X(i)T+12γW(i))−1X(i)X(i)T(X(i)X(i)T+12γW(i))−T−(X(i)X(i)T+12γW(i))−1−(X(i)X(i)T+12γW(i))−T]X(i)+I+L(i).

Step 3: Updating αi while fixing P(i) and Y. The optimal solution of problem Eq. (36) can be calculated as:

(42) αi=[1/tr(YG(i)YT+P(i)TW(i)P(i))](1/(r−1))∑i=1v[1/tr(YG(i)YT+P(i)TW(i)P(i))](1/(r−1)).

We alternatively update P(i)T, Y and αi until convergence. The complete procedures are described in Algorithm 2.

Algorithm 2 Algorithm of RMLPRE.

Input: Given a set of multi-view datasets X(i)∈Rd(v)×n, the number of iterations T, the paramenter r, T and γ, and the embedded space dimensions m.	
Initialization: αi=1/v, W(i)∈Rn×n;	
Compute N by Eq. (32);	
  for t=1:T do	
    Compute Y according to Eq. (41);	
    Update P(i) according to Eq. (38);	
    Update W(i) according to Eq. (39);	
    Update αi according to Eq. (42);	
  end for	
Output: The contribution weight αi, and the projection matrix P(i).	

Time complexity analysis

The time complexity of Algorithm 1 is mainly determined by the computational costs of its key equations. Assuming

(43) dmax=max{d(i)|i=1,2,…,v}.

For Algorithm 1, the time complexity of Eqs. (31), (34), (35) and (29) are O(n∗dmax2+dmax3+n2∗dmax), O(n3), O(n2+m∗n2+n∗m2), O(n∗dmax2+dmax3+m∗n∗dmax), respectively. Given that m≪d(i), the main time complexity of Algorithm 1 is O(dmax3+n∗dmax2+n2∗dmax+n3). Similarly, for Algorithm 2, the operations and their associated costs are comparable, resulting in the main time complexity of O(dmax3+n∗dmax2+n2∗dmax+n3).

Experiments

To validate the effectiveness of our proposed frameworks, we compared the classification accuracy with traditional multi-view methods, including LPCCA (Sun & Chen, 2007), ALPCCA (Wang & Zhang, 2013), MCCA (Rupnik & Shawe-Taylor, 2010), DMCCA (Gao et al., 2018), MvDA (Shu et al., 2019), MvPLS (Cao et al., 2018), EWNMF (Wei et al., 2023), MUNPE (Jayashree, Shiva Prakash & Venugopal, 2024). Experiments were conducted on three real-world datasets using a Windows 10 desktop computer with a 2.5 GHz Intel Core i5-7300HQ CPU, 64 GB of RAM, and Matlab R2019b (64-bit).

Datasets description

Coil (https://cave.cs.columbia.edu/repository/COIL-20): The Coil Dataset (Nene, Nayar & Murase, 1996), originating from Columbia University, contains a diverse collection of 1,400 images, featuring 20 different objects. Each object is represented by a substantial set of 72 images, offering rich variability for analysis.

Orl (http://www.cl.cam.ac.uk/research/dtg/attarchive/facedatabase.html): Hailing from the Olivetti Laboratory in Cambridge, England, the ORL Dataset (Samaria & Harter, 1994) comprises 400 images showcasing the faces of 40 distinct individuals. These images were captured under a range of conditions, including differing lighting, positions, and expressions.

Yale (https://vision.ucsd.edu/datasets/yale-face-database): The Yale Face Dataset contains 165 images of 15 people’s faces, with variations in lighting, expression, and posture.

Our multi-view graph embedding method is designed to process data in vector form. For example, for the image data, we first preprocess it to extract numerical vector features of gray-scale intensity (GSI), local binary patterns (LBP), and histogram of oriented gradients (HOG), respectively. Then, the proposed method, as well as comparison methods, is applied to extract the features of these numerical vector representations. For some details, please refer to Table 1. As shown in Fig. 2, the image features extracted using the three described techniques display significant differences, representing three distinct perspectives of the image data. Furthermore, we evaluated the robustness of our proposed methods by introducing salt-and-pepper noise with densities of 0.1 and 0.3 to each dataset.

Table 1 Experimental details about datasets.

Dataset	Views	Noise	No. classes	Samples	Training samples	Features	
Coil	3 (GSI, LBP, HOG)	None, 0.1, 0.3	20	1,440	1,152	4,096, 4,096, 1,764	
Orl	3 (GSI, LBP, HOG)	None, 0.1, 0.3	40	400	320	2,000, 2,000, 720	
Yale	3 (GSI, LBP, HOG)	None, 0.1, 0.3	15	165	132	2,000, 2,000, 720	

Figure 2 Image features extracted using three techniques on the Coil, ORL and Yale Face datasets.

Experiments setup

In our experiments, we employed a random split of the data, allocating 80% of the samples for training and reserving the remaining 20% for testing. The number of training samples for each dataset is provided in Table 1. For multi-view datasets {X1,…,Xv}, once we obtained the projection matrices {P1,…,Pv}, we extracted embedding features for each view as follows:

(44) Y(i)=P(i)TX(i),i=1,…,v.

We applied a 1NN classifier for classification, and these experiments were repeated five times. The evaluation criteria were based on the average classification accuracies of the embedding representations. Optimal parameters were determined through grid search with γ,λ∈{2−5,2−3,2−1,2,23,25}. The parameters of the algorithms employed for comparison were rigorously adhered to as delineated within the original publication.

Experiment results

On the real-world datasets

In our results, we commence by presenting the classification accuracy results across feature dimensions ranging from 10 to 90 for each dataset. The summarized results for all methods can be found in Tables 2–4, with the best performance highlighted in bold. Furthermore, we provide a visual representation of the classification accuracy of all methods under various reduced dimensions for each dataset in Fig. 3. Based on our experimental findings, several key observations are noted.

Table 2 1NN classification accuracy (%) for Coil Dataset.

Bold entries indicate the best results.

	10	20	30	40	50	60	70	80	90	
PCA (Oh & Kwak, 2016)	9.03	9.14	9.18	9.20	9.17	9.19	9.20	9.20	9.21	
LDA (Belhumeur, Hespanha & Kriegman, 1997)	48.53	49.42	49.52	49.63	49.66	49.67	49.68	49.69	49.69	
ALPCCA (Wang & Zhang, 2013)	18.77	32.69	44.30	53.13	58.39	62.90	65.26	66.40	66.45	
LPCCA (Sun & Chen, 2007)	41.81	44.13	45.82	46.04	46.47	46.82	47.18	47.45	47.38	
MCCA (Rupnik & Shawe-Taylor, 2010)	46.04	59.38	65.39	69.65	72.03	73.78	75.21	76.21	77.04	
DMCCA (Gao et al., 2018)	67.11	72.46	74.73	75.84	76.31	77.21	70.12	74.69	78.29	
MvDA (Shu et al., 2019)	53.80	50.76	45.02	39.03	40.06	41.51	42.60	43.91	44.73	
MvPLS (Cao et al., 2018)	44.06	44.89	45.13	45.08	44.98	44.80	44.65	44.64	44.56	
EWNMF (Wei et al., 2023)	69.23	71.57	72.78	73.95	74.23	73.38	74.61	76.93	77.27	
MUNPE (Jayashree, Shiva Prakash & Venugopal, 2024)	68.83	70.86	67.75	67.11	66.67	64.86	65.42	63.61	61.81	
pcMRE	80.11	82.58	81.93	80.71	78.97	76.52	74.79	73.17	71.74	
pcMLPRE	80.05	82.52	81.91	80.72	78.91	76.51	74.69	73.03	71.59	
pcRMLPRE	78.83	80.94	80.40	80.03	79.34	78.64	78.13	78.08	78.44	
daMRE	85.44	86.83	87.30	86.09	84.67	82.01	79.50	76.93	74.89	
daMLPRE	85.94	86.84	87.21	86.41	85.37	84.08	83.65	83.41	83.23	
daRMLPRE	84.46	86.53	84.12	81.73	80.32	79.65	78.96	78.44	78.45	

Table 3 1NN classification accuracy (%) for ORL dataset.

Bold entries indicate the best results.

	10	20	30	40	50	60	70	80	90	
PCA (Oh & Kwak, 2016)	8.80	9.38	9.43	9.43	9.50	9.45	9.50	9.50	9.53	
LDA (Belhumeur, Hespanha & Kriegman, 1997)	28.38	29.13	29.38	29.63	29.63	29.63	29.63	29.75	29.75	
ALPCCA (Wang & Zhang, 2013)	10.42	18.33	26.04	32.10	38.29	44.58	49.79	52.56	55.92	
LPCCA (Sun & Chen, 2007)	23.38	35.94	41.42	43.08	45.40	48.15	50.67	53.65	56.25	
MCCA (Rupnik & Shawe-Taylor, 2010)	15.63	27.50	38.75	45.50	50.83	57.67	61.92	66.17	68.92	
DMCCA (Gao et al., 2018)	55.50	68.25	74.92	78.67	80.04	81.29	83.33	84.29	85.63	
MvDA (Shu et al., 2019)	56.96	56.33	60.17	53.75	45.08	45.17	44.79	44.67	45.21	
MvPLS (Cao et al., 2018)	41.58	44.54	45.71	46.08	46.21	46.33	46.04	46.08	46.13	
EWNMF (Wei et al., 2023)	65.50	73.16	81.86	85.20	86.47	87.19	87.41	87.85	88.26	
MUNPE (Jayashree, Shiva Prakash & Venugopal, 2024)	49.58	55.00	55.83	58.33	60.00	55.42	60.00	59.58	54.17	
pcMRE	68.15	81.83	86.08	88.50	88.08	89.42	88.58	88.67	87.67	
pcMLPRE	68.17	82.00	86.08	88.42	87.92	89.17	88.67	88.83	87.75	
pcRMLPRE	72.58	83.25	85.08	86.83	87.67	89.33	89.42	88.58	87.75	
daMRE	80.79	86.17	86.97	90.91	92.42	89.17	89.92	90.83	90.00	
daMLPRE	80.83	86.25	86.92	90.83	91.75	88.58	88.33	89.75	89.42	
daRMLPRE	80.17	87.42	89.33	94.17	92.08	90.42	87.75	86.25	84.92	

Table 4 1NN classification accuracy (%) for Yale Face Database.

Bold entries indicate the best results.

	10	20	30	40	50	60	70	80	90	
PCA (Oh & Kwak, 2016)	7.20	7.96	8.13	8.27	8.31	8.27	8.44	8.49	8.49	
LDA (Belhumeur, Hespanha & Kriegman, 1997)	21.56	22.22	22.22	22.22	22.22	22.22	22.22	22.22	22.22	
ALPCCA (Wang & Zhang, 2013)	31.52	50.78	60.33	65.59	69.11	71.37	72.26	73.85	73.70	
LPCCA (Sun & Chen, 2007)	24.56	30.48	35.74	38.30	40.74	42.41	43.56	44.52	45.37	
MCCA (Rupnik & Shawe-Taylor, 2010)	43.33	56.74	63.04	66.44	69.41	70.22	71.48	72.74	72.89	
DMCCA (Gao et al., 2018)	80.30	87.63	92.74	94.67	89.56	80.30	73.85	75.04	76.37	
MvDA (Shu et al., 2019)	36.30	38.81	39.04	40.30	40.30	40.44	40.59	40.59	40.74	
MvPLS (Cao et al., 2018)	36.15	39.33	40.00	40.07	40.22	40.07	40.44	40.74	41.26	
EWNMF (Wei et al., 2023)	63.33	74.64	79.47	82.64	83.75	83.96	84.10	84.61	84.97	
MUNPE (Jayashree, Shiva Prakash & Venugopal, 2024)	35.56	38.52	39.26	41.48	37.04	39.26	33.33	35.46	37.78	
pcMRE	75.46	82.87	89.93	93.63	92.00	89.48	88.44	86.96	84.30	
pcMLPRE	75.85	82.96	89.78	93.48	92.59	89.93	89.48	87.85	85.78	
pcRMLPRE	85.78	87.70	90.67	92.44	91.41	90.22	90.07	89.93	88.89	
daMRE	94.81	96.15	92.59	95.70	98.52	98.07	97.48	96.74	96.44	
daMLPRE	94.96	98.37	96.30	96.30	98.48	98.93	98.07	97.19	95.70	
daRMLPRE	94.67	95.70	93.78	91.11	89.48	88.85	87.19	86.56	85.48	

Figure 3 Classification accuracies with a different dimension.

(A) Coil Dataset. (B) Orl dataset. (C) Yale Face Database.

The tables and figure unequivocally illustrate that, in nearly all instances, our proposed methods achieve superior classification accuracy compared to the comparison method. Specifically, in the realm of supervised methods, the MLPRE-based methods typically attain the highest accuracy, often peaking around 50 dimensions. However, it is notable that the classification accuracy curve does not always maintain a high level as dimensionality reduction dimension increases. This behavior is primarily due to the F-norm regularization term’s inclination to retain more feature information, and with higher dimensions, redundant information becomes more prominent, adversely affecting model accuracy. In sum, when using the F-norm regularization term, it is advisable to aim for a dimension reduction of around 50.

In the unsupervised methods, the proposed RMLPRE-based methods usually achieve the highest classification accuracy and tends to be more stable compared to the MLPRE-based methods. This stability is attributed to the L2,1-norm regularization, which tends to select a small number of features close to zero, rendering it more robust. In addition, based on the classification results, methods that incorporate shared structural information items generally outperform those that lack these items. This superiority is because the MLPRE-based methods generally outperform the MRE-based methods in terms of classification accuracy.

Considering the classification results, it is evident that different norm regularization terms have distinct applicable ranges. The L2,1-norm regularizer is more suitable for unsupervised methods, while the F-norm regularizer is better suited for supervised methods.

On the real-world datasets with added noise

Thereafter, our experiments extended to verify the robustness of the proposed methods. We initially present the classification accuracy results across feature dimensions from 10 to 90 for each dataset with added noise. The comprehensive results for all methods are detailed in Tables 5–10, with the optimal performance highlighted in bold. We also offer a visual representation of the classification accuracy of all methods at various reduced dimensions for each dataset in Figs. 4 and 5. Based on our experimental findings, several key observations emerge.

Table 5 1NN classification accuracy (%) for Coil Dataset with 0.1 salt-pepper noise added.

Bold entries indicate the best results.

	10	20	30	40	50	60	70	80	90	
PCA (Oh & Kwak, 2016)	7.37	7.25	6.99	6.58	6.25	5.96	5.84	5.76	5.63	
LDA (Belhumeur, Hespanha & Kriegman, 1997)	11.84	11.96	11.21	11.07	10.99	11.24	11.23	11.36	10.62	
ALPCCA (Wang & Zhang, 2013)	18.88	27.61	34.79	40.40	44.12	44.60	44.32	45.36	46.83	
LPCCA (Sun & Chen, 2007)	7.61	7.28	8.74	8.70	8.42	8.18	7.43	6.48	7.16	
MCCA (Rupnik & Shawe-Taylor, 2010)	14.16	20.05	26.25	31.54	34.31	37.11	39.50	41.71	42.93	
DMCCA (Gao et al., 2018)	52.21	64.44	67.16	69.72	71.78	73.09	29.92	29.90	30.01	
MvDA (Shu et al., 2019)	69.93	35.51	31.98	32.14	31.90	31.97	31.81	31.98	31.99	
MvPLS (Cao et al., 2018)	41.18	41.98	42.08	41.41	40.87	40.17	39.83	39.21	38.82	
EWNMF (Wei et al., 2023)	36.72	45.75	49.75	55.76	57.92	58.26	59.73	61.29	62.74	
MUNPE (Jayashree, Shiva Prakash & Venugopal, 2024)	47.28	50.36	50.33	49.72	47.39	47.97	46.97	46.33	45.58	
pcMRE	61.93	67.62	65.77	62.76	61.20	57.62	55.69	54.92	52.43	
pcMLPRE	66.18	70.71	69.82	68.36	66.67	64.88	63.91	62.67	63.56	
pcRMLPRE	64.35	70.68	67.16	67.52	66.99	67.39	66.98	66.61	65.86	
daMRE	70.91	76.29	81.16	77.74	75.25	72.38	70.03	68.93	68.51	
daMLPRE	70.96	76.37	80.49	79.05	76.78	74.63	71.39	70.34	67.73	
daRMLPRE	69.02	76.93	77.09	71.04	68.58	65.88	63.41	64.76	67.50	

Table 6 1NN classification accuracy (%) for ORL Dataset with 0.1 salt-pepper noise added.

Bold entries indicate the best results.

	10	20	30	40	50	60	70	80	90	
PCA (Oh & Kwak, 2016)	6.43	6.98	7.23	7.30	7.23	7.33	7.28	7.13	6.93	
LDA (Belhumeur, Hespanha & Kriegman, 1997)	5.38	4.75	5.00	4.50	4.63	4.63	4.50	4.38	4.50	
ALPCCA (Wang & Zhang, 2013)	8.10	12.96	17.15	20.79	24.00	26.54	28.13	29.31	30.73	
LPCCA (Sun & Chen, 2007)	9.46	9.44	9.85	10.67	9.19	10.42	11.96	12.90	13.63	
MCCA (Rupnik & Shawe-Taylor, 2010)	6.83	9.25	12.25	15.92	17.92	20.63	21.88	24.33	27.88	
DMCCA (Gao et al., 2018)	24.42	37.63	45.25	51.88	50.13	50.21	50.67	55.75	58.38	
MvDA (Shu et al., 2019)	61.83	76.67	83.46	31.67	30.71	30.71	29.38	28.88	29.21	
MvPLS (Cao et al., 2018)	34.92	40.08	42.00	42.00	42.71	42.63	42.50	42.08	42.29	
EWNMF (Wei et al., 2023)	45.48	53.92	56.76	58.94	62.34	61.88	64.66	65.18	65.79	
MUNPE (Jayashree, Shiva Prakash & Venugopal, 2024)	22.08	27.50	33.75	25.00	30.00	28.33	30.42	31.25	31.25	
pcMRE	46.58	60.33	63.25	65.25	66.25	66.75	66.42	66.58	67.92	
pcMLPRE	60.42	68.83	70.58	69.42	68.92	68.00	66.92	66.75	66.00	
pcRMLPRE	55.83	66.33	68.00	65.42	68.42	68.42	68.83	70.42	68.67	
daMRE	68.42	70.56	74.83	81.03	85.67	83.42	81.75	78.08	76.50	
daMLPRE	68.33	70.58	74.77	81.00	85.83	83.92	83.08	80.58	80.17	
daRMLPRE	61.25	66.67	68.67	79.83	83.08	76.25	72.58	69.08	66.42	

Table 7 1NN classification accuracy (%) for Yale Face Database with 0.1 salt-pepper noise added.

Bold entries indicate the best results.

	10	20	30	40	50	60	70	80	90	
PCA (Oh & Kwak, 2016)	6.62	7.73	8.22	8.13	8.18	8.04	7.51	7.82	7.96	
LDA (Belhumeur, Hespanha & Kriegman, 1997)	13.56	13.11	13.56	14.00	13.56	13.33	13.33	13.56	13.78	
ALPCCA (Wang & Zhang, 2013)	21.33	34.93	40.89	45.41	48.59	49.11	51.56	51.22	52.30	
LPCCA (Sun & Chen, 2007)	21.41	24.96	26.89	25.48	23.19	23.37	24.63	26.78	27.81	
MCCA (Rupnik & Shawe-Taylor, 2010)	19.04	24.74	29.63	32.37	37.04	40.44	42.00	41.93	43.56	
DMCCA (Gao et al., 2018)	58.74	62.89	70.52	75.33	27.26	28.67	27.63	27.11	28.00	
MvDA (Shu et al., 2019)	85.41	33.11	32.89	32.59	32.22	31.85	32.00	31.56	32.22	
MvPLS (Cao et al., 2018)	31.70	35.56	37.85	38.15	39.70	39.78	40.44	39.70	39.56	
EWNMF (Wei et al., 2023)	42.86	51.49	57.94	60.09	65.12	65.63	71.83	75.93	78.19	
MUNPE (Jayashree, Shiva Prakash & Venugopal, 2024)	22.22	31.85	25.93	31.11	33.33	34.81	34.81	37.78	33.33	
pcMRE	56.30	68.30	70.22	68.44	66.37	68.59	70.22	74.22	77.78	
pcMLPRE	66.52	78.22	78.22	79.56	77.93	76.59	74.96	72.89	66.67	
pcRMLPRE	65.63	74.96	77.33	78.33	78.81	77.15	75.74	76.48	76.22	
daMRE	82.01	91.26	90.07	90.22	88.44	85.78	82.96	78.00	73.85	
daMLPRE	82.07	90.22	89.15	89.75	88.11	87.41	87.56	87.26	87.81	
daRMLPRE	67.26	81.48	78.11	78.26	77.96	79.00	73.96	70.33	69.44	

Table 8 1NN classification accuracy (%) for Coil Dataset with 0.3 salt-pepper noise added.

Bold entries indicate the best results.

	10	20	30	40	50	60	70	80	90	
PCA (Oh & Kwak, 2016)	7.34	6.85	6.31	5.88	5.42	5.30	5.03	5.04	5.04	
LDA (Belhumeur, Hespanha & Kriegman, 1997)	8.19	8.40	8.38	8.33	8.27	8.50	8.47	8.50	8.55	
ALPCCA (Wang & Zhang, 2013)	14.41	19.98	23.02	24.15	24.43	24.07	24.93	24.60	25.26	
LPCCA (Sun & Chen, 2007)	5.20	5.33	6.17	6.31	6.28	6.34	5.53	5.37	5.54	
MCCA (Rupnik & Shawe-Taylor, 2010)	6.96	9.27	10.48	10.93	11.97	12.79	14.02	14.37	15.74	
DMCCA (Gao et al., 2018)	28.03	37.82	39.11	41.05	44.04	45.94	22.07	21.83	22.07	
MvDA (Shu et al., 2019)	51.23	28.76	27.31	27.24	26.90	26.67	26.59	26.50	26.59	
MvPLS (Cao et al., 2018)	34.41	32.66	31.74	30.73	30.00	28.99	28.31	28.38	28.06	
EWNMF (Wei et al., 2023)	23.86	31.85	32.05	32.17	33.42	33.62	33.71	34.14	34.27	
MUNPE (Jayashree, Shiva Prakash & Venugopal, 2024)	27.33	31.11	30.47	30.25	30.31	29.53	28.89	29.06	28.31	
pcMRE	39.94	38.87	40.15	41.25	38.95	41.74	41.11	39.33	39.01	
pcMLPRE	44.59	44.84	45.28	44.39	43.43	41.24	39.20	37.46	35.87	
pcRMLPRE	40.98	32.61	31.27	29.71	28.54	28.63	27.67	27.61	27.19	
daMRE	47.86	52.77	66.46	63.26	57.53	52.38	51.13	50.27	50.49	
daMLPRE	47.88	52.83	66.94	60.12	53.81	49.51	46.66	45.71	44.76	
daRMLPRE	45.64	51.24	54.49	47.54	39.57	37.04	35.83	32.85	33.09	

Table 9 1NN classification accuracy (%) for ORL Dataset with 0.3 salt-pepper noise added.

Bold entries indicate the best results.

	10	20	30	40	50	60	70	80	90	
PCA (Oh & Kwak, 2016)	6.50	6.85	6.38	6.33	6.33	5.93	6.08	6.03	5.88	
LDA (Belhumeur, Hespanha & Kriegman, 1997)	4.38	4.25	4.50	4.13	4.38	4.25	4.87	4.50	4.75	
ALPCCA (Wang & Zhang, 2013)	4.38	5.48	5.92	6.65	7.10	7.06	7.27	6.77	7.04	
LPCCA (Sun & Chen, 2007)	3.67	3.85	4.38	4.08	3.10	4.60	4.81	5.27	5.29	
MCCA (Rupnik & Shawe-Taylor, 2010)	3.42	4.17	5.04	5.96	6.08	6.21	6.54	6.88	6.75	
DMCCA (Gao et al., 2018)	11.17	16.21	18.58	21.50	19.21	17.50	18.92	20.75	22.25	
MvDA (Shu et al., 2019)	29.08	39.29	50.42	19.29	17.83	17.17	16.21	16.38	16.21	
MvPLS (Cao et al., 2018)	22.79	25.38	23.25	23.42	23.13	22.96	22.67	22.79	22.58	
EWNMF (Wei et al., 2023)	6.67	9.72	11.43	12.05	12.43	13.85	12.94	14.36	14.67	
MUNPE (Jayashree, Shiva Prakash & Venugopal, 2024)	10.42	9.17	10.42	8.75	9.58	9.17	7.08	8.75	10.42	
pcMRE	14.42	15.50	15.08	15.67	18.00	19.08	19.67	22.33	23.42	
pcMLPRE	16.75	18.67	18.67	20.08	20.42	19.17	21.08	19.92	20.92	
pcRMLPRE	6.83	8.75	9.25	10.00	11.00	11.67	12.42	12.33	12.50	
daMRE	37.17	36.50	42.67	53.39	53.50	49.17	47.33	47.08	46.08	
daMLPRE	37.47	36.52	42.77	53.42	54.00	49.33	44.83	41.58	40.67	
daRMLPRE	25.92	24.67	28.33	30.08	29.83	26.42	23.67	22.83	22.92	

Table 10 1NN classification accuracy (%) for Yale Face Database with 0.3 salt-pepper noise added.

Bold entries indicate the best results.

	10	20	30	40	50	60	70	80	90	
PCA (Oh & Kwak, 2016)	5.82	5.91	6.09	6.18	6.27	6.18	6.40	6.40	6.84	
LDA (Belhumeur, Hespanha & Kriegman, 1997)	10.00	7.78	8.44	9.78	10.44	10.00	10.22	10.22	10.89	
ALPCCA (Wang & Zhang, 2013)	10.11	12.59	14.07	16.00	16.81	17.41	18.11	17.41	17.56	
LPCCA (Sun & Chen, 2007)	12.19	13.15	12.81	16.26	14.15	14.74	16.44	14.93	14.93	
MCCA (Rupnik & Shawe-Taylor, 2010)	8.59	11.85	11.04	12.59	12.00	13.26	13.41	14.74	15.26	
DMCCA (Gao et al., 2018)	25.41	25.93	29.70	32.52	14.37	13.85	13.93	14.59	13.48	
MvDA (Shu et al., 2019)	48.81	18.96	17.93	18.37	17.78	17.70	17.56	18.15	18.15	
MvPLS (Cao et al., 2018)	22.81	23.56	22.30	22.44	22.81	23.33	24.30	25.78	24.52	
EWNMF (Wei et al., 2023)	19.46	21.75	22.30	21.36	22.96	23.33	24.54	24.75	25.03	
MUNPE (Jayashree, Shiva Prakash & Venugopal, 2024)	14.07	8.15	8.15	8.89	10.37	7.41	5.93	8.89	8.89	
pcMRE	20.59	30.22	31.41	30.52	28.30	32.30	33.63	36.44	35.11	
pcMLPRE	30.67	30.22	27.56	26.22	27.26	27.11	26.52	25.63	25.04	
pcRMLPRE	17.19	19.56	21.19	21.19	20.15	21.33	21.63	20.15	21.33	
daMRE	48.62	70.22	62.52	57.04	51.85	56.30	51.70	48.74	48.85	
daMLPRE	48.59	71.70	59.26	49.33	41.04	36.15	34.78	33.85	33.66	
daRMLPRE	34.52	34.22	33.56	35.85	31.70	30.22	30.07	31.85	31.89	

Figure 4 Classification accuracies with a different dimension (0.1 salt-pepper noise).

(A) Coil Dataset. (B) ORL Dataset. (C) Yale Face Database.

Figure 5 Classification accuracies with a different dimension (0.3 salt-pepper noise).

(A) Coil Dataset. (B) ORL Dataset. (C) Yale Face Database.

Regarding Tables 5–7 and Fig. 4, the following conclusions can be made:

The proposed methods continue to perform well even with small amounts of noise in the data. Methods that incorporate shared structural information terms (MLPRE-based methods) generally outperform those without these terms (MRE-based methods) across the board. The L2,1-norm regularization remains more suitable for unsupervised methods, while the F-norm regularizer remains better suited for supervised methods.

According to Tables 8–10 and Fig. 5, the following conclusions can be made:

With increasing levels of noise, the accuracy and stability of various methods decline, but our methods still show significant advantages. The L2,1-norm regularization methods (RMLPRE-based methods) exhibit significantly lower accuracy compared to the F-norm, suggesting that the L2,1-norm regularizer is less robust in high-noise data. Interestingly, methods incorporating shared structural information terms (MLPRE-based methods) start to perform worse than those without these terms (MRE-based methods). This is due to the severe inaccuracy of the shared k-nearest neighbour in a high-noise environment, which negatively affects the performance of the models.

Experimental results on both real and noisy datasets demonstrate that our framework effectively captures the consistency and complementarity of multi-view data, with robustness further enhanced by incorporating L2,1-norm constraints. Moreover, by employing a nonlinear shared embedding, the framework significantly alleviates the critical information loss commonly associated with linear projection methods. This approach enables better retention of discriminative information, ensuring stable performance even in noisy conditions and underscoring the advantages of our framework over traditional methods.

Ablation study

The effects of each component of the model framework on the feature extraction results are evaluated through ablation studies.

Table 11 presents the peak accuracy of the k-nearest neighbors algorithm following dimensionality reduction through different combinations of components within the model framework on datasets with varying noise levels. The results reveal that the norm term of the projection matrix exerts the most pronounced influence on the discriminative capability of feature extraction outcomes, with the F-norm term showing a slight advantage over the L2,1-norm term. In the context of low-noise datasets, the shared k-nearest neighbor structure term has relatively little influence on the discriminative power of feature extraction. However, it significantly improves the accueacy of subsequent classification algorithms in noisy conditions.

Table 11 Experimental results of ablation study.

Dataset	GE (PCA)	∥P(i)TX(i)−Y∥F2	∥P(i)T∥F2	∥P(i)T∥2,1	WjkLP(i)∥Yj−Yk∥F2	Accuracy (%)	
Coil	✓	✓	✗	✗	✗	24.19	
	✓	✓	✓	✗	✗	82.58	
	✓	✓	✗	✓	✗	79.17	
	✓	✓	✗	✗	✓	24.76	
	✓	✓	✓	✗	✓	82.52	
	✓	✓	✗	✓	✓	80.94	
Coil (0.1 salt-pepper noise)	✓	✓	✗	✗	✗	5.38	
	✓	✓	✓	✗	✗	67.62	
	✓	✓	✗	✓	✗	62.72	
	✓	✓	✗	✗	✓	5.42	
	✓	✓	✓	✗	✓	70.71	
	✓	✓	✗	✓	✓	70.68	
Coil (0.3 salt-pepper noise)	✓	✓	✗	✗	✗	5.77	
	✓	✓	✓	✗	✗	41.74	
	✓	✓	✗	✓	✗	21.18	
	✓	✓	✗	✗	✓	6.96	
	✓	✓	✓	✗	✓	45.28	
	✓	✓	✗	✓	✓	40.98	

By minimizing the norm of the projection matrix, we strive to obtain a “simplified” projection matrix, which preserves the structural integrity of the original data after dimensionality reduction, thus preventing structural imbalances caused by overfitting of the training dataset and improving the discriminant ability of feature extraction. Regarding the simplification of the projection matrix, the F-norm term is more inclined to select an abundance of features near zero, whereas the L2,1-norm term results in a sparser set of feature vectors, exhibiting lower sensitivity to outliers. The superiority of the F-norm term over the L2,1-norm term in this research is attributed to the linear projection fitting term’s inherent outlier rejection, which reduces the effectiveness of the L2,1-norm term. As a result, in the ablation studies, the L2,1-norm term failed to outperform the F-norm term in influencing feature extraction discriminability. The global shared k-nearest neighbor structure term is beneficial for preserving the spatial structure of the data. In low-noise datasets, where data is rich in information, the linear projection fitting term and the projection matrix norm term are sufficient to maintain the intrinsic structural relationships, rendering the global shared k-nearest neighbor structure term less impactful. Conversely, in high-noise scenarios where the original data information is heavily distorted, the global shared k-nearest neighbor structure term significantly augments the discriminative ability of feature extraction by complementing the intrinsic structural relationships.

Influence of parameters

To explore the impact of parameters γ and λ on model performance, we conducted experiments by extracting the features to 50 dimensions and then classifying them. We’ve created corresponding curves depicting the average classification accuracy as a function of these parameters, as shown in Fig. 6. Based on the figures, we’ve made the following conclusions:

Figure 6 The parameters influence.

(A) pcMRE. (B) pcMLPRE. (C) pcRMLPRE. (D) daMRE. (E) daMLPRE. (F) daRMLPRE.

In the MRE framework, pcMRE appears to be relatively insensitive to the parameter γ, while daMRE is more sensitive to this parameter. The highest classification accuracy is typically achieved when γ=2. In the MLPRE framework, pcMLPRE is generally insensitive to the parameter γ. When γ is less than or equal to 2, pcMLPRE is insensitive to the parameter λ. However, when γ is greater than 2, pcMLPRE becomes sensitive to the parameter λ, and the average classification accuracy increases with an increase in this parameter. Conversely, daMLPRE is sensitive to the parameter γ, and the average classification accuracy decreases as this parameter increases. daMLPRE is also sensitive to the parameter λ, with the highest classification accuracy typically occurring when λ=2. In the RMLPRE framework, pcRMLPRE is sensitive to both parameters γ and λ. The classification accuracy tends to be higher when λ=0.5. Similarly, daRMLPRE is sensitive to both parameters γ and λ, with the highest classification accuracy typically occurring when λ=8.

In summary, the choice of parameters γ and λ significantly influences the classification accuracy of various models and frameworks. Fine-tuning these parameters allows us to identify the optimal configurations for achieving the best performance under different conditions. Therefore, careful parameter selection is a crucial consideration during the experimental process, as it can substantially enhance a model’s classification accuracy, better aligning it with the requirements of the downstream tasks.

Convergence analysis

We uniformly extract the features to 50 dimensions using our methods. Figure 7 shows the convergence curves of the proposed methods. As depicted in the figure, the objective function value for the MRE and MLPRE methods remains nearly constant after the third iteration. Similarly, for the RMLPRE method, it stabilizes after seven iterations. This demonstrates that our proposed methods converge rapidly, with the MRE and MLPRE methods converging faster than the RMLPRE method.

Figure 7 Convergence curves.

(A) pcMRE. (B) pcMLPRE. (C) pcRMLPRE. (D) daMRE. (E) daMLPRE. (F) daRMLPRE.

Conclusions

This article introduces three novel multi-view feature extraction frameworks based on regression embedding, extending single-view graph embedding models to the multi-view scenario. Our frameworks meticulously consider the consistency and complementarity of multi-view data, emphasizing robustness to noisy datasets. Furthermore, the utilization of non-linear shared embedding helps prevent the loss of essential information that can occur with linear projection techniques. Through numerical experiments, we verify the effectiveness and robustness of our frameworks on both real and noisy datasets. In particular, we analyze the applicability, advantages, and disadvantages of each framework, providing a solid theoretical and experimental basis for choosing the appropriate framework for specific tasks.

However, it is important to note that our frameworks only extend the single-view graph embedding methods, and they may not be applicable to some other excellent single-view methods. In our future work, we aim to develop a more general framework that acts as a bridge between single-view feature extraction and multi-view feature extraction.

Supplemental Information

Supplemental Information 1 This file includes raw data and code.

Additional Information and Declarations

Competing Interests

Author Contributions

Data Availability

The authors declare that they have no competing interests.

Ling Jing conceived and designed the experiments, performed the experiments, analyzed the data, performed the computation work, prepared figures and/or tables, authored or reviewed drafts of the article, and approved the final draft.

Yi Li conceived and designed the experiments, performed the experiments, performed the computation work, prepared figures and/or tables, and approved the final draft.

Hongjie Zhang conceived and designed the experiments, performed the experiments, analyzed the data, performed the computation work, authored or reviewed drafts of the article, and approved the final draft.

The following information was supplied regarding data availability:

The experimental data and code for each dataset are available in the Supplemental Files.

The Coil Dataset is available at https://cave.cs.columbia.edu/repository/COIL-20.

Our Database of Faces, formerly ‘The ORL Database of Faces’, is available at: http://www.cl.cam.ac.uk/research/dtg/attarchive/facedatabase.html.

The Yale Face Database is available at: https://vision.ucsd.edu/datasets/yale-face-database.

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
