# Peer review of "Robust multi-view locality preserving regression embedding"

_PeerJ Computer Science, doi:10.7717/peerj-cs.2619_

## Round 0.1 · original submission · Major Revisions

Dear authors,

Thank you for submitting your article. Feedback from the reviewers is now available. It is not recommended that your article be published in its current format. However, we strongly recommend that you address the issues raised by the reviewers, especially those related to readability, experimental design and validity, and resubmit your paper after making the necessary changes. Before submitting the paper following should also be addressed:

1. All of the values for the parameters of the algorithms selected for comparison should be provided.
2. Please pay special attention to the usage of abbreviations. Spell out the full term at its first mention, indicate its abbreviation in parenthesis and use the abbreviation from then on.
3. Equations should be used with correct equation number. Many of the equations are part of the related sentences. Attention is needed for correct sentence formation.

Best wishes,

Reviewer 1 ·

Basic reporting

well done

Experimental design

Rich experimental design.

Validity of the findings

The paper has a certain degree of innovation and clear conclusions.

Additional comments

The paper presents three innovative multi-view feature extraction frameworks based on regression embedding. The paper is well-organized. However, I have several suggestions that could improve the clarity and impact of the manuscript.
1. There are minor grammatical errors and awkward phrasings throughout the manuscript. I recommend a thorough proofreading by a native English speaker or a professional editor.
2. the time complexity of given algorithmis should be analysis.
3. Although the paper presents experimental results, the analysis and discussion of the results are not in-depth. It is suggested that the authors further discuss the mechanisms behind the experimental results and the advantages and disadvantages of the proposed method compared to existing methods.

Cite this review as

Reviewer 2 ·

Basic reporting

The paper proposes a multi-view regression embedding framework for the feature extraction domain. The proposed method considers the consistency and complementarity issues of multiview data, enhances robustness by incorporating paradigm constraints, and proposes to avoid loss of key information with nonlinear shared embedding, and finally performs a performance evaluation on a real dataset. Reading the full paper, I have the following questions:
1.Graph Embedding in the narrow sense is to represent the relationships of nodes, edges, etc. in the graph structure with numerical vectors. And image embedding represents embedding image information into low-dimensional vector space. The expression of graph embedding is used throughout the text, while the experiments use picture data, which is easy to produce ambiguity.
2.The abstract does not describe the advantages of multiple views, which could be introduced appropriately.
3.It is hoped that the author will discuss the connection and application significance between other graph-based methods and the methods proposed in this manuscript
4.Some of the references in the text can be shown in the form of corner notation. In many places the insertion of references interferes with reading.
5.Whether the references and most comparisons methods are too old.
6.The pictures in the text can be embellished.
7.Have ablation experiments been performed on the nonlinear part of the structure?
8.Are PCA and LDA traditional GE models?
9.Check the expression of English in the paper, e.g. 'r is a positive parameter ....'.
10.Check the expression and consistency of letter symbols, punctuation marks in formulas in the paper.
11.Have you thought about replacing the adaptive weighting parameter with an attention mechanism.
12.You can show a comparison of the multi-view data constructed with GSI, LBP, and HOG methods under different datasets
13. At the end of page 11, a large number of pictures block the text in the paragraph.
14.Are comparison experiments performed with traditional PCA or LDA methods? Comparison methods can be simply adding or splicing raw data from multiple views to synthesize single view data.
15.The multiple views constructed with GSI, LBP, HOG methods are not real data. In neural networks, it is possible to construct multiple views from a single view by the above methods and splice them into different channels as input data. In principle, neural networks have realized the concept of multi-view fusion feature extraction.
16.The object of your research is images, so what is the application side of your research in today's world where image processing is developing so rapidly. Will it be replaced by adaptive feature extraction algorithms like neural networks?
17.The object of your research is image embedding, and the traditional Transformer architecture is also capable of image embedding. May I ask what is the practicality and specificity of your research?
18.If your research includes other data types, then describe the data type scenarios to which your method applies and conduct the experiments.
19.The comparison methods in all tables require references

Experimental design

no comment

Validity of the findings

no comment

Additional comments

no comment

Cite this review as

---

## Round 0.2 · accepted · Accept

Dear Author,

Thank you for clearly addressing the reviewer's comments. Your paper seems now sufficiently improved after the last revision. Your manuscript is ready for publication.

Best wishes,

Reviewer 1 ·

Basic reporting

The paper is clearly expressed in English. References are more comprehensively cited. Figures and tables specifications.The conclusion is correct.

Experimental design

Experimental design is reasonably adequate. Comparative analysis with multiple algorithms.

Validity of the findings

The paper is innovative. Changes have been made to the comments made.

Cite this review as

Reviewer 2 ·

Basic reporting

Compared with the previous version, the manuscript has some improvements, but there are still many grammatical errors, such as: Image missing on page 12, line 359. and line 425.

Experimental design

N/A

Validity of the findings

N/A

Additional comments

N/A

Cite this review as